# New insights into US flood vulnerability revealed from flood insurance big data

Oliver E.J. Wing [1,2✉], Nicholas Pinter[3,4], Paul D. Bates [1,2] & Carolyn Kousky[5]

Improvements in modelling power and input data have vastly improved the precision of physical flood models, but translation into economic outputs requires depth–damage functions that are inadequately verified. In particular, flood damage is widely assumed to increase monotonically with water depth. Here, we assess flood vulnerability in the US using >2 million claims from the National Flood Insurance Program (NFIP). NFIP claims data are messy, but the size of the dataset provides powerful empirical tests of damage patterns and modelling approaches. We show that current depth–damage functions consist of disparate relationships that match poorly with observations. Observed flood losses are not monotonic functions of depth, but instead better follow a beta function, with bimodal distributions for different water depths. Uncertainty in flood losses has been called the main bottleneck in flood risk studies, an obstacle that may be remedied using large-scale empirical flood damage data.

[1] School of Geographical Sciences, University of Bristol, Bristol, UK. [2] Fathom, Bristol, UK. [3] Department for Earth and Planetary Sciences, University of California, Davis, CA, USA. [4] Center for Watershed Sciences, University of California, Davis, CA, USA. [5] Wharton Risk Center, University of Pennsylvania, Philadelphia, PA, USA. ✉email: oliver.wing@bristol.ac.uk

Flooding is the deadliest and most costly natural disaster, both in the US and worldwide, with global damages exceeding $1 trillion since 1980[1]. Both climate change and continued development of flood-prone areas may enhance these losses by up to a factor of 20 by the end of the century[2]. Numerous studies have focused on quantifying future flood losses at scales from local to global; all of which involve a translation from the physical phenomenon of flooding (e.g., extent and depth of inundation) to its economic impacts (e.g., dollars of damage). This translation normally requires relationships between water depth and the resulting asset damages, referred to as depth–damage functions or curves. Depth–damage curves typically stipulate loss (structural and/or content damage, in total or as a percentage of structure value) as a monotonic function of inundation depth. Different curves are commonly applied for different occupancy classes (e.g., single-family residential vs. commercial) and for a variety of construction types. Standard depth–damage functions have been produced for different geographical regions, for example, in the United Kingdom in the so called Multi-Coloured Manual[3] and the United States which has curves compiled by the U.S. Army Corps of Engineers[4]. Many researchers and practitioners use these relationships off-the-shelf, assuming they are well-calibrated and universally applicable. In fact, a wealth of literature notes the substantial scatter and underappreciated uncertainty in depth–damage estimates[5–8]. As Freni et al.[9] summarized, uncertainty derived from depth–damage curves is the main bottleneck in estimating flood damage for a wide variety of applications ranging from climate change studies to cost-benefit calculations justifying massive infrastructure projects.

By contrast, flood hazard modelling techniques have undergone revolutionary advancements in recent years, owing to exponential growth in computational capacity and of high-resolution topographic and other spatial data. These developments have heralded a step-change in our understanding of flood hazard, even at continental and global scales[2,10–12]. However, the output required for most studies is dollars (or other currency units) of flood losses—for applied and planning studies in particular—and the translation of hydrology to economic loss requires passage through a depth–damage function. For example, even a hydraulic model based on state-of-the-science lidar topography and multibeam bathymetry and calibrated to within millimeters may yield damage estimates off by an order of magnitude after translation through the depth-damage bottleneck[13].

Most applied studies of the economic impacts of flooding rely upon existing depth–damage relationships. In the US, widely used curves have been compiled by the Army Corps of Engineers (USACE) and/or by the Federal Insurance Administration (FIA; now Federal Insurance and Mitigation Administration). These depth–damage functions are typically maintained in tabular format, specifying structure and content damages as a percentage of structure value for a given water depth (usually in 1-foot [0.3 m] increments) above or below the first occupied level of the structure. FIA and the USACE iteratively developed these curves beginning in 1970 using early NFIP loss data, local Corps studies, and the collective judgment of experts[14]. These US federal depth–damage functions are employed universally in flood risk assessments nationally (e.g. in FEMA's HAZUS-MH software[15]) and often globally. We are unaware of any systematic empirical verification these curves, although Skaggs and Davis[14] noted that the USACE was (in 1993) "exploring the possibility of using FIA's massive claims database" to interrogate these damage functions.

Previous attempts to verify the reliability of depth–damage curves have been limited by data availability. What scarce data is available is often summarized at a high degree of spatial aggregation (e.g., municipal or regional level), while specific testing of

the vulnerability component of risk models requires structure-by-structure information. Further, even building-level losses can only be used to evaluate vulnerability if the observed damage is reported alongside observed hazard intensity such as local water depth. Thus, vulnerability data that are spatially aggregated or lack corresponding individual hazard intensity measures (e.g., flood depth) require modeled representations of hazard and exposure before losses can be calculated. Many studies, therefore, have compared how the use of different vulnerability models impact damage estimates using modeled hazard and exposure, but have not analyzed the validity of depth–damage curves per se[5,16–19]. Other smaller scale studies have employed detailed data and found considerable variability in the relationship between inundation depth and damage[20,21]. This has led to a proliferation of multivariate models from which, for example, flood damage is predicted as a function of water velocity, duration and contamination; the timing, quality, and nature of early warnings; precautions and preventative measures applied; and/or building quality and socio-economic status of residents[21–23]. These multivariate models provide valuable information to better understand structural fragility and societal resilience to flooding, yet such complex approaches can be difficult to apply in standard flood risk assessments.

Despite this growing recognition that there is weak correlation between depth and damage, the use of monotonic depth–damage functions in risk analyses remains widespread. This runs contrary to literature which suggests the treatment of depth–damage functions should be probabilistic[7–9,21–24]. We thus employ the National Flood Insurance Program's (NFIP) database of over 2M historic flood claims to provide empirical insights into building vulnerability in the US, where we find low agreement between NFIP damage observations and commonly applied depth–damage approaches. Instead, the true stochastic distribution of depth–damage can be better characterized by a beta distribution, where proportional damage to buildings is generally concentrated at minimal and maximal loss. Increased water depth shifts the beta distribution, whereby the likelihood of maximal (minimal) damage increases (decreases). We further find vulnerability is highly heterogeneous both within and between US states, and we reveal that floodplain management practices have reduced vulnerability within specified risk areas but increased it outside of them.

## Results

**Nationwide analysis of depth–damage**. We analyzed 2,085,015 NFIP flood damage claims (1972–2014) from a database obtained directly from FEMA (see Data Availability) that includes fields for structural damages, structure value, depth of inundation above the first occupied level, and many other categories (see Methods). Of the 2,085,015 total claims, 976,363 records contained all of the information (depth, damage, structure type and value) needed to assess damage dependence on water depth. The database here differs from claims data recently released by FEMA[25], which lack additional explanatory fields (e.g. depth) necessary for analyses of this kind.

The NFIP claims data include a number of clear artefacts. Most notably, depths should be in units of feet, but instead, in some cases, seem to be recorded in inches. Anomalous numbers of claims are recorded with depths of 12 and 24 feet and, to a lesser extent, with higher multiples of 12 (also purportedly feet). Looking at the overall claim frequency per unit water depth, interpolation of numbers of claims at intuitive inch values (e.g. 6, 12, 18, 24) suggests that ~5% of the total database is impacted by this issue. For example, for the purported depth values of 6 feet—an intuitive value for an insurance adjuster to select as half of 1 foot

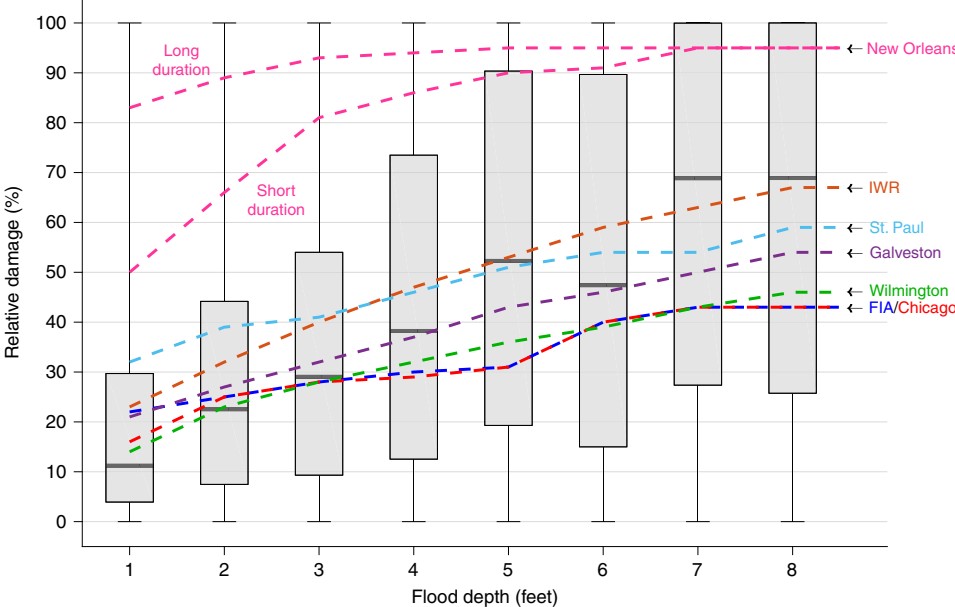

**Fig. 1 Comparison of NFIP-derived and US federal depth–damage relationships.** The boxplots show the distribution of NFIP-derived relative structure damages (structural damages as a percentage of building value) at each flood depth increment for residential buildings with one story and no basement. Minima and maxima are shown at the foot of the whiskers, 25th and 75th percentiles bound the shaded box with medians (50th percentile) within. Dashed lines represent select US depth–damage curves for this type of building; labels represent the USACE District or other source of these functions. These curves relate to freshwater flooding; the New Orleans District has a separate curve for both long and short duration flooding.

(6 inches)—the trend of increasing damage with increasing depth is interrupted (Fig. 1). In this case, we cannot discern the damages caused by 6 feet of water, since a subset of 6 inch records seem to be present within this class. Despite this challenge, the benefit of such a large empirical dataset is that robust trends can be identified, and the noise generated by erroneous records is dwarfed by the actual depth-dependence. As a result, in this study, we restrict our analysis to low (<8) integer values of depth where records in feet seem to dominate.

Structural damage to residential buildings with one story and no basement constitute a majority (493,707, or 51%) of the NFIP claims analyzed here. For simplicity, we focus on this building type so sample sizes remain large across all subsequent analyses. Median relative damages (damage/structure value) to this building type (Fig. 1) increase monotonically with increasing water depth above the base of the first occupied floor (1–8 feet), excepting the value at 6 feet (a likely data artefact; see above), however, the variability about these medians is extensive. For instance, the median damage at 1-foot of flood inundation is 11% of building value, yet the central 50% of claims range from 4% ($Q_{25}$) to 30% ($Q_{75}$) damage. At 5-feet depth, the $Q_{25}$–$Q_{75}$ range is 19–90% damage. Single depth–damage curves lose both the breadth and the nature of this variability. Figure 1 also shows eight commonly used curves, compiled by the USACE or FIA, which describe the vulnerability of one-storey residential structures without a basement to freshwater flooding. These functions appear internally disparate (even for similar types of structure and flooding) and do not represent either the central tendencies or variability of depth–damage according to the empirical records in the NFIP claims database.

In fact, the NFIP claims data show that such relative flood damages do not fit, and cannot be described well, by any central-tendency distribution. The NFIP-derived distributions of damage for each inundation depth increment (Fig. 2) are distinctly bimodal—with a disproportionate number of claims at both the high and low ends of the relative damage spectrum. At shallower

inundation depths, a plurality of claims lie in the 0–10% relative damage bin. This is consistent with previous analyses of NFIP claims that found that the majority of insured flood damages in the US were relatively small, with a median value of $12,555 (1980–2012; single-family residences)[26]. In contrast, reasons why a disproportionate number of properties suffer near complete losses, even at relatively small inundation depths remain unclear, and this question merits further detailed research. One potential explanation may be the development of mold in flooded properties which could lead to a total write-off of the structure. With increasing depths, low relative damages account for a progressively lower proportion of total claims, while the number of claims within the 90–100% damage bin increases.

Thus, relative flood damages for individual depths are not well fit by a median, mode, or any other central-tendency value. Instead, we find that relative flood damages for each depth increment within the NFIP claims database is better described by a beta distribution (Supplementary Fig. 1). The beta distribution seems to be applicable to losses for a number of different natural hazards, and the insurance industry has been modelling vulnerability this way for a number of years[27]. Whereas the USACE and other depth–damage curves consist of presumed central-tendency damages that increase monotonically with increasing flood depths, the NFIP-based losses are much better characterized by a family of beta functions in which the controlling parameters (e.g., α and β in Supplementary Fig. 1) cause the bimodal beta distribution to shift and swing toward greater damages as depth increases. We illustrate the ability of the beta distribution to represent the shape and scale of the depth–damage relationship (i.e., Fig. 2) in Supplementary Note 1. Supplementary Figs. 2 and 3 illustrate the degree of fit between the beta and empirical distributions ($R^2$ values in Supplementary Table 1), particularly when compared to the one-to-one depth–damage approach embodied in the US federal curves.

Further, we examine the accuracy of the US curves by using them to estimate the damages in the claims data (given inundation depth and structure value) and comparing these

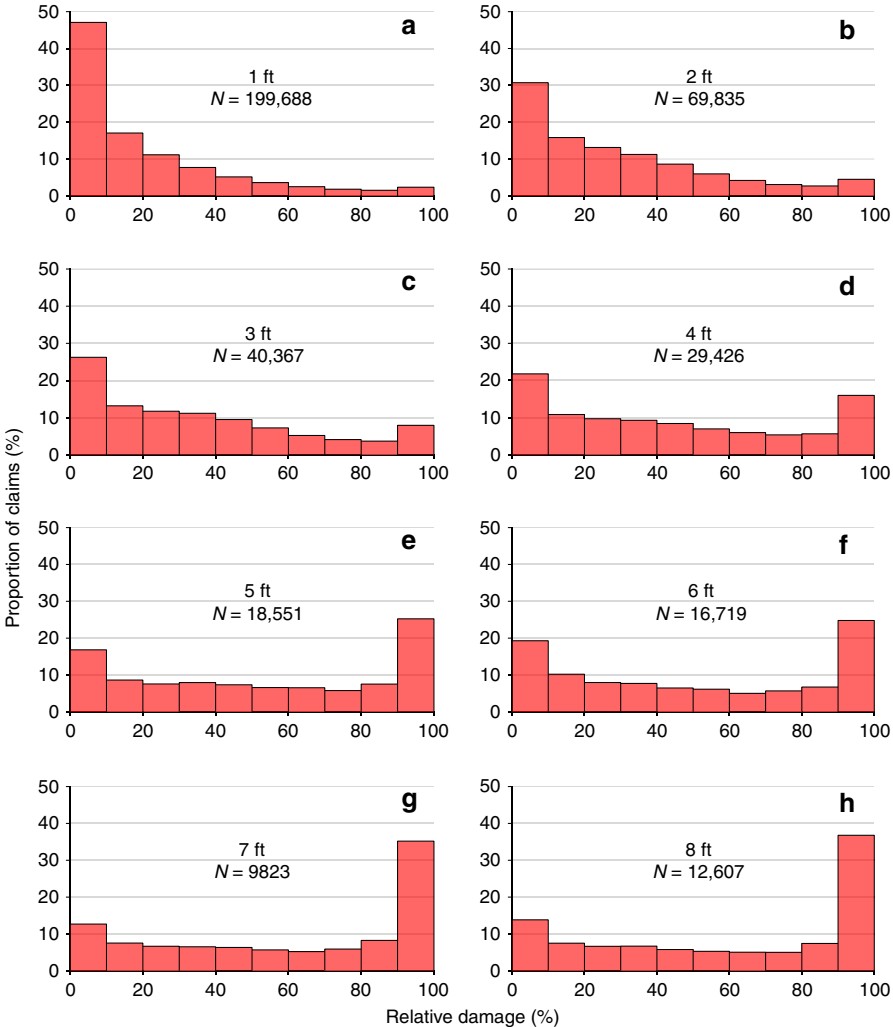

**Fig. 2 The shape of NFIP-derived depth–damage distributions.** Each panel shows the proportion of observations which fall within a relative flood damage bin (structural damages as a percentage of building value, binned in increments of 10% between 0 and 100%) for each inundation depth increment from 1 to 8 feet **a**–**h** from the NFIP claims database. Totals here are for single-family residential buildings with one story and no basement.

estimates to the recorded damage (Supplementary Table 2). In this calculation, we use the FIA curve; the most commonly used of the federal depth–damage functions[4]. Patterns in Fig. 1 were generally reflected in these results too: in aggregate, shallow depth–damage was overestimated by ~25%, while deep depth–damage was underestimated by ~25%. We used the coefficient of determination to measure the predictive power of the FIA curve, which does so by balancing the sum of squared errors (residual variance) and the total sum of squares (data variance); see Eq. (1). Across all depth increments, the coefficient of determination was negative, meaning that the damage associated with each claim is better predicted by the mean observed damage than it is by using the FIA curve. That is, the residual variance is greater than the data variance. Mean absolute building-level errors in damage estimation were 84% of the mean recorded damage across all depth increments. At 1-foot inundation, this error rose to 105% of the mean damage.

Realistic estimations of economic flood losses thus require both a probabilistic treatment of flood vulnerability as well as an accurate characterization of the stochastic relationship between inundation depth and the resulting damages—neither of which are generally captured in the large majority of current applications.

**Regional patterns in depth–damage.** Some of the broad variability in flood damages illustrated in Figs. 1 and 2 could plausibly be the result of systematic geographical variation in vulnerability; for example, in typical construction type, prevailing repair costs, and other factors that may vary in different regions of the US. In traditional applications of depth–damage curves (in the US at least), this variation is recognized and adjusted by the use of curves developed by different regional Districts of the USACE. For example, the St. Paul District, covering Minnesota and Wisconsin, stipulates that 1-foot of water causes 32% relative structural damage to a one-story residence. In contrast, the USACE Chicago District (northeast Illinois and northwestern Indiana) curves suggest just 16% damage for the same 1 foot of inundation—i.e., half the level of damage predicted for nearby Wisconsin and Minnesota.

In order to test regional variations in depth–damage across the US, we subdivided the NFIP claims data by zip code and by state (Fig. 3). For regions with existing USACE District depth–damage curves, we compared those curves (as shown in Fig. 1) with the empirical NFIP claims for the corresponding geographical area. These comparisons show that, for example, the characterization of the St. Paul District as more vulnerable (higher flood fragility) than the Chicago District (Fig. 1) is not justified; in fact, damages

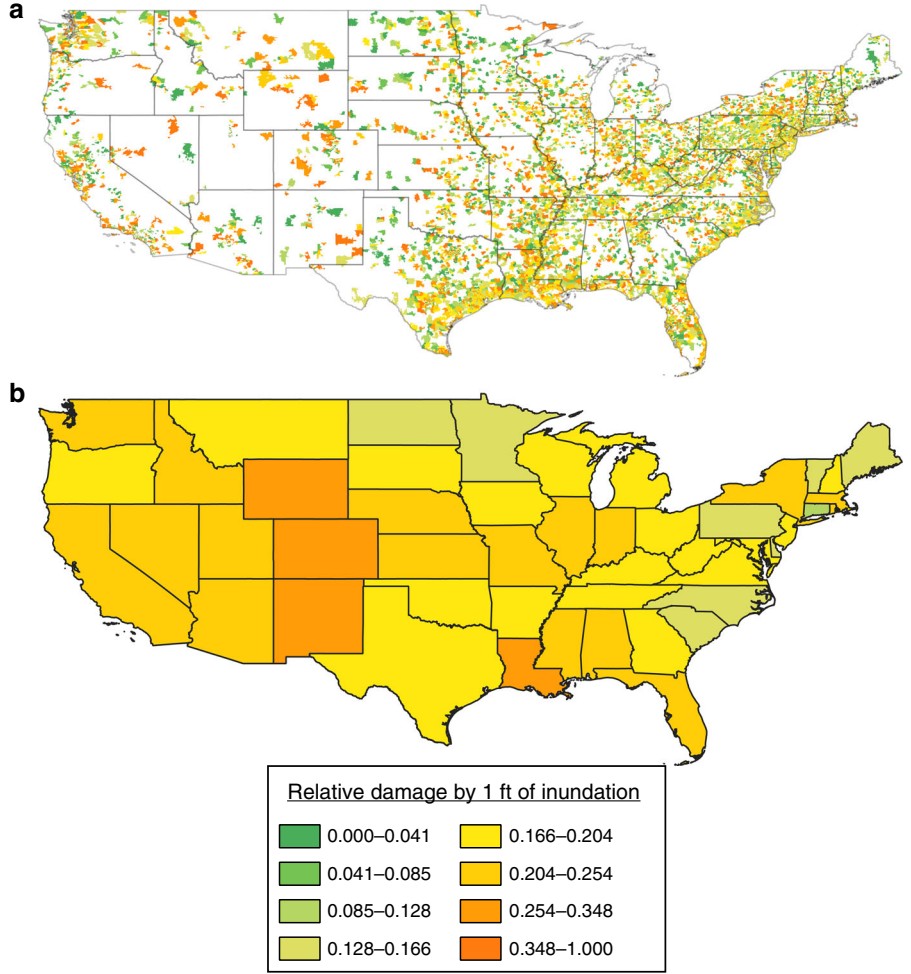

**Fig. 3 The impact of 1 foot of inundation across the US.** These maps show the spatial distribution of relative structural damages arising from NFIP claims where a one-story building without a basement was inundated by 1-foot of floodwater. Each spatial unit is representative of the mean damage within its area, summarized: **a** by zip code tabulation area and **b** by state. Blank areas in **a** indicate where no claims data exist to generate a depth–damage relationship.

at 1-foot inundation are generally lower in the St. Paul District (16%) than in the Chicago District (24%). Even for regional USACE functions where the empirical data qualitatively support the differences in the curves, local variations in flood losses (see Fig. 3a) are far greater than similarities and differences at the state or regional level. For example, broadly lower damages (16% at 1-foot inundation) in the Wilmington District covering North Carolina (Fig. 3b) mask considerable within-district variation between zip codes (1–87% damage at 1-foot). Similarly, while the Galveston curve and the mean NFIP damage within the District both stipulate 21% damage at 1-foot inundation, zip code-level vulnerability varies between 2 and 74%.

To further examine the regional USACE curves, we isolated claims arising from flooding in Chicago in July 1996 (Chicago District), Tropical Storm Allison in June 2001 (Galveston District), Hurricane Ike in September 2008 (Galveston District), and Hurricane Isabel in September 2003 (Wilmington District). Using NFIP structure values and inundation depths, the corresponding USACE District's depth–damage function was applied to estimate losses (Supplementary Table 3). Total event damages were generally well replicated by the USACE curves: for example, Hurricane Ike losses were estimated at $447.0 million vs. $450.0 million in recorded NFIP losses. However, this aggregate match was a fortuitous balance that includes large over- and under-prediction errors for individual claims (Supplementary

Fig. 4). For instance, for Hurricane Ike damages, the USACE curves underpredicted individual losses by $284 on average, yet the spread of errors about this mean was large: mean absolute error was $34,213, or ~75% of the mean recorded damage for Ike of $45,264. For all four events, the coefficient of determination was negative, meaning building-level damages are better predicted by the mean observed damage than by the USACE curve. The reasonable match in event damage totals amidst this extremely low predictive capability of USACE functions for individual losses suggests that regional curves may have been calibrated to major flood events in aggregate, and not for single structures, as is commonly understood.

Regional patterns in the NFIP claims data are difficult to discern in Fig. 3. There seems to be an east–west divide in depth-damage dependence. Areas west of the Mississippi River appear to be more vulnerable to 1 foot of flood inundation than areas to the east. The western US is generally more arid than the east, so that when flood-producing precipitation does occur—one might speculate— damages may be greater as buildings may not be constructed with flood resilience in mind. Vulnerability in coastal areas may be driven by claims arising from saltwater flooding, which is generally more damaging for a given depth than freshwater flooding[3]. Some NFIP flood claims were specified as being due to freshwater or saltwater flooding, but the resulting relative damages were fairly insensitive to flood type (Supplementary Fig. 5). As such, we do not

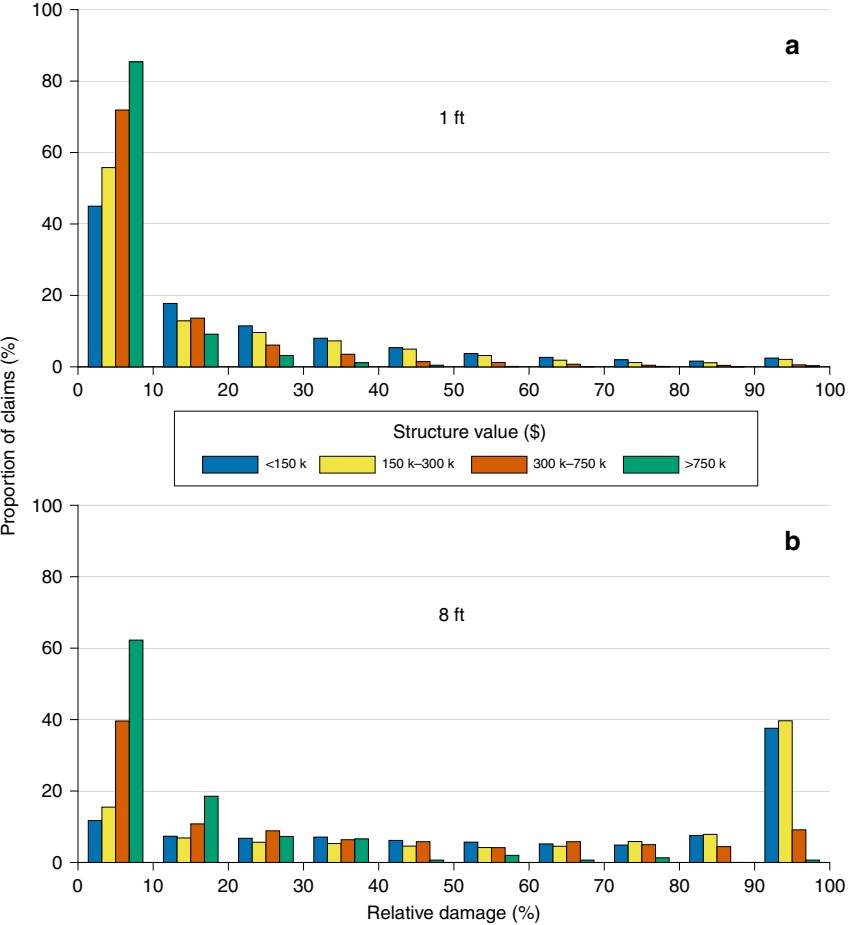

**Fig. 4 Relative damage to one-story, no-basement residential homes of different values.** Sample water depths are illustrated: 1 foot (**a**) and 8 feet (**b**). Damages are binned in increments of 10%: e.g., 62% of claims relating to structures of value >$750,000 where 8-feet of inundation occurred incurred between 0 and 10% relative damages.

separate fluvial and coastal flood claims. The states with the highest flood fragility in Fig. 3b are Louisiana, New Mexico, Colorado, and Wyoming, but these states individually have very different patterns and histories of flooding. NFIP claims in Louisiana are singularly driven by losses during Hurricane Katrina in 2005. Claims in Colorado seem to be dominated by losses during flash flooding along the Front Range in 2013. In contrast, New Mexico and Wyoming have experienced few loss events over the history of NFIP. Clearly, these data should be analyzed carefully, in a rigorous multivariate and multilevel context, but as a whole, current assumptions regarding the geography of vulnerability based on USACE curves do not seem to match the empirical data.

**Depth–damage dependence on structure value**. We analyzed relative damages as a function of the value of each structure (Fig. 4). These structure values are recorded in the NFIP database as total assessed value, and differ in many cases from the insured value, which is subject to a regulatory cap. Unsurprisingly, the relative damage-dependence on total structure value is shown most clearly for the proportion of claims for the lowest increment, 0–10% of each structure's value. For 1 foot of inundation, buildings valued >$750,000 incurred 0–10% damage for 85% of claims; for buildings worth <$150,000, the proportion of 0–10% damage claims is just 45%. In other words, less expensive homes are more likely to experience greater relative damage, presumably because of fixed costs such as for building materials and labor. This pattern strengthens for progressively deeper depths. For example, for 8 feet

of inundation, homes valued >$750,000 experienced 0–10% damage 62% of the time, compared to just 12% for homes valued <$150,000. Looking at damages approaching structures' total value (90–100%), at 8-feet inundation, <1% of claims fell into this category for >$750,000 homes versus 38% for <$150,000 homes. The differences illustrate that absolute flood damages are, to some extent, value-independent, or at least not directly proportional to the value of a structure.

**Depth–damage dependence on age and flood zone**. The average age of residential building stock in a region may impact the degree of damages experienced during a major flood. Newer homes may be built to higher standards, but alternatively may be more costly to repair. Similarly, areas with more recent residential development should have a greater proportion of structures built after the first FEMA flood hazard maps (see discussion below) and thus (hopefully) meet minimum NFIP regulations, most notably construction in the 100-year floodplain above base flood elevation (the elevation floodwaters are anticipated to reach during a 100-year event). We tested the hypothesis that newer residential structures would experience generally lower levels of damage for a given water depth. As a whole, and on average, the NFIP claims data support this hypothesis (Supplementary Fig. 6). For example, at 5 feet of flood inundation, homes built after 1980 incurred minimal (0–10%) relative structural damage 22% of the time, compared to just 12% of claims for pre-1950 buildings. The same pattern is seen for shallower inundation. At 1 foot of

inundation, 52% of claims on post-1980 buildings incurred 0–10% loss, compared to ~43% of claims for pre-1980 structures. Generally, for a given inundation depth, newer buildings were more likely to experience minimal relative losses (0–10%) and less likely to experience catastrophic relative losses (90–100%) than older buildings. Given that these relationships are for equal inundation depths (i.e., the elevation of the building with respect to base flood elevation is irrelevant), it appears that newer buildings withstand flooding better than older buildings.

The NFIP database also records whether claims were on pre-FIRM or post-FIRM structures, meaning constructed before or after the first Flood Insurance Rate Map (FIRM) in that jurisdiction. Post-FIRM structures comprise roughly 90% of structures built after 1980 in the claims database. Because owners of pre-FIRM structures may have been unaware of their flood risk at the time of construction or purchase, NFIP policies on most of these structures historically have been granted subsidized premiums substantially below actuarial rates. In addition, pre-FIRM policies include unmitigated structures on FEMA floodplains that, at least theoretically, should not be present among post-FIRM policies. When pre-FIRM and post-FIRM claims are further separated as within the FEMA 100-year floodplain vs. properties outside the floodplain (Fig. 5), interesting contrasts emerge. Post-FIRM homes constructed within the flood zone, meaning with recognition of flood risk, are systematically more likely to experience low (0–10%) damages than pre-FIRM structures. At 1-foot of inundation, 51% of post-FIRM claims within the flood zone incurred 0–10% structural damage, compared to 43% for pre-FIRM properties. This suggests that some degree of resilience has been added to flood-zone properties.

Outside the 100-year flood zone, the pattern reverses, and claims on pre-FIRM residential structures are more likely to incur minimal damage (0–10%) compared to post-FIRM structures. At 1-foot of inundation, pre-FIRM properties outside the flood zone experience 0–10% damage 47% of the time, compared to 42% for post-FIRM properties. Furthermore, post-FIRM homes outside the floodplain tend to have a higher susceptibility to catastrophic (90–100%) damage than pre-FIRM buildings. For example, at 8-feet of inundation, 52% of post-FIRM buildings experience 90–100% damage outside the flood zone, compared to only 33% of pre-FIRM buildings. Over the history of the NFIP, at least 27% of flood claims and 14% of claim dollars have been for properties outside FEMA's 100-year floodplain (A zones and coastal V zones). Some of these claims are due to shallow and distributed drainage flooding; others are likely due to shortcomings of FEMA mapping across the US[28–30]. But more broadly, whereas post-FIRM homes on floodplains seem to have mitigated at least some of their flood risk, newer residences constructed outside FEMA flood zones are significantly more susceptible, not less, to flood damage. For all depth increments studied here, the portion of post-FIRM claims of 0–10% damage is less for structures situated outside the FEMA 100-year floodplain than for structures within it, and the portion of post-FIRM claims of maximal (90–100%) damage is greater outside the mapped floodplain than within it. Publication of FEMA maps may lead to complacency for those who fall outside of designated flood zone. Since FEMA FIRMs denote flood risk largely as binary—you are either at risk or not—those who fall into the 'not' category may be less prepared and therefore experience greater losses.

The binary nature of flood mapping in the US has been extensively criticized[31,32]. Problems with binary flood mapping include challenges to accurately specifying the 100-year flood limit; for example, data constraints such as the limited length and availability of streamflow records. In addition, climate change, human modifications of rivers and floodplains, and other sources of hydrologic non-stationarity have shifted, and are shifting, flood probabilities in ways not currently accounted for in FEMA mapping[33,34]. Furthermore, FEMA mapping only covers roughly 60% of contiguous US land area and often does not capture flooding on small streams or from intense localized rainfall[11]. More localized analyses have documented the impact of binary zonation on flood risk; particularly, how it impairs effective mitigation activities outside the specified risk zone[35,36]. Further, the binary system has been observed to cause a clustering of development in the unregulated areas around the boundary of the risk zone, leading to significant 'off-floodplain' losses[37,38]. Analysis of nationwide flood insurance claims adds further weight to the conclusions of previous studies: that the US should consider transitioning to a program where flooding is contoured and communicated beyond single-probability, in-or-out flood risk.

## Discussion

A wide variety of applications currently rely upon flood depth–damage curves that provide a single damage value corresponding to a given inundation depth for each structure type (e.g., one-story residential). Those damage values are assumed to be central tendencies that are representative of the underlying variability and increase monotonically for all increments of additional water depth. A detailed examination of empirical losses in the large NFIP claims database suggests that those assumptions may be invalid, and that flood risk quantified from existing depth–damage functions may significantly mischaracterize flood vulnerability at the level of individual structures and often more broadly. While a monotonic relationship is intuitive for flooding of a single structure—where each additional increment of flood depth should add incrementally to damage to that structure—for large populations of flooded structures, the monotonic function does not apply. Using the NFIP claims, damage instead can be described by a beta distribution, where most losses are concentrated at the low and the high extremes of the relative damage spectrum. With each incremental increase in inundation depth, the beta distribution shifts and rolls towards the maximal-damage mode. This variability in depth–damage is not adequately described by any central tendency. For example, if 100 buildings were flooded to a depth of 4 feet, assuming that all 100 uniformly incur the median of 38% damage is very different from a bimodal or beta distribution in which 22 buildings experienced <10% damage, 16 buildings experienced >90% damage, and the remaining 62 something in between (10–90%). Plans for effective disaster preparedness, response, recovery, and mitigation all require a correct understanding of these relationships[39]. Indeed, many billions of dollars of infrastructure investment and risk management decisions in the US are based on cost-benefit analyses that employ existing depth–damage curves. In general, these curves overestimate damages induced by shallow-water flooding—which comprises the large majority of losses—and generally underestimate damages by deep inundation, compared to the NFIP data.

Looking forward, large empirical datasets, such as the NFIP claims analyzed here, could be the basis for a multivariate, probabilistic approach to flood damage estimation that explicitly captures the stochasticity in depth–damage. Geographic variations in vulnerability (east/west divide, in/out of the FEMA flood zone), building age (as well as pre/post FIRM), structure value, and other fields not explored here (e.g., wall and foundation type) may aid in the refinement of more realistic functions, ultimately leading to flood risk assessments of increased validity. Such data also shed light on the effectiveness of floodplain management

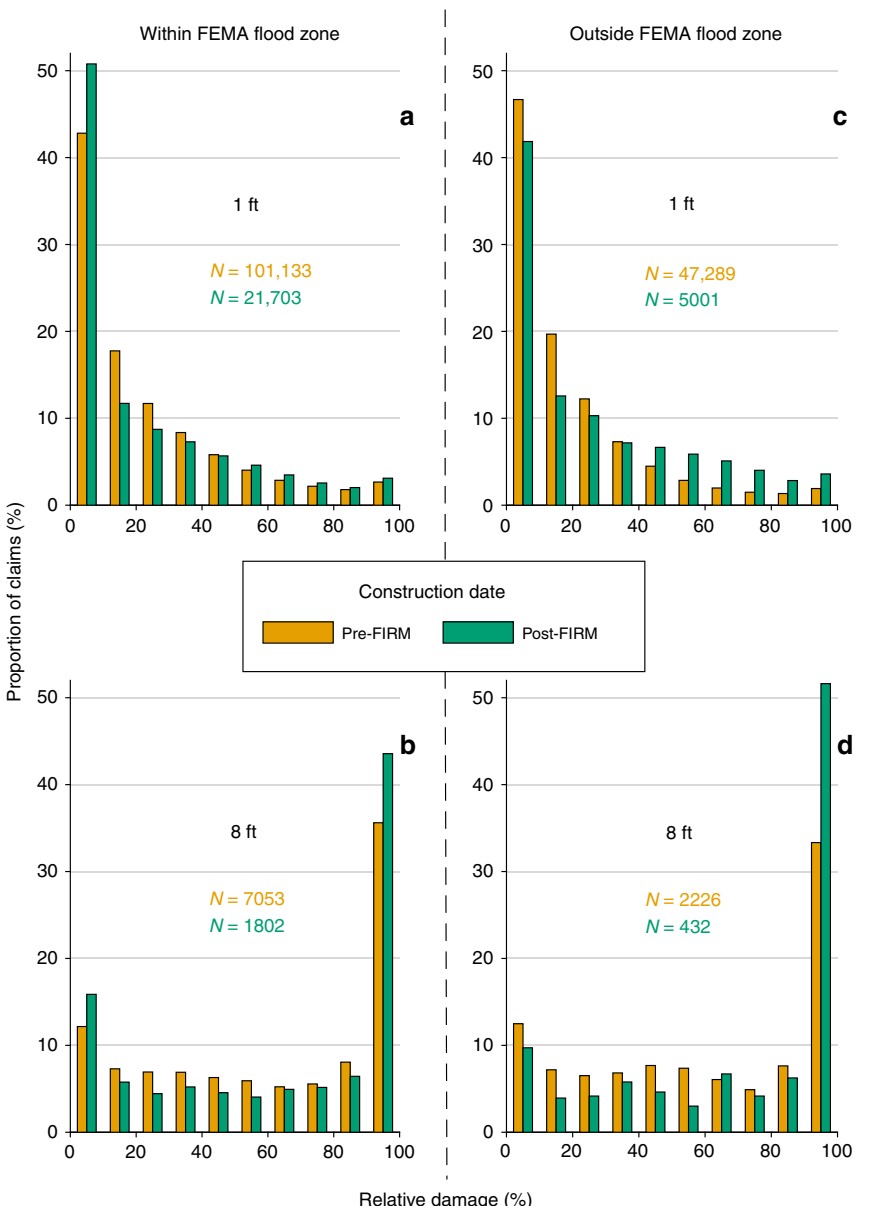

**Fig. 5 Relative damage to buildings that are situated within vs. outside of FIRMs before vs. after their initial publication.** Selected results are shown for claims arising from one-story residential structures with no basement within the FEMA 100-year flood zones for 1 (**a**) and 8 (**b**) feet of inundation and the same but for properties outside of the FEMA flood zone for 1 (**c**) and 8 (**d**) feet of inundation.

decisions in the US. While post-FIRM properties within the FEMA 100-year flood zone appeared more resilient to flooding than pre-FIRM properties, this phenomenon switches outside of the 100-year flood zone. This observation further highlights the fallacy of binary risk zonation.

We have entered an era where industry and governments are increasingly looking for solutions from big data, meaning large and complex data sets from which predictive lessons can be learned through rigorous analysis. In June 2019, FEMA publicly released a version of its >2 million NFIP claims records, with identifiable property locations and most other data fields removed[25]. The full data set, a portion of which was analyzed here, contains rich information for managing flood risk across the US and beyond. The power of large empirical data sets is that their volume can reveal patterns even when obscured by overlapping independent variables and data error and uncertainty. We encourage FEMA to share a larger portion of its data, including

more precise location information, and allow the research and actuarial communities to better measure, map, and model present and future flood risk.

## Methods

**National Flood Insurance Program claims database**. We analyzed 2,085,015 flood damage claims from the U.S. National Flood Insurance Program (NFIP), over the period 1972 to 2014. This confidential database was obtained directly from FEMA and includes fields for total structural damages, value of the structure, and depth of inundation above the first occupied floor as recorded by a loss adjuster. A number of entries for individual structures in the database are blank due to incomplete data; others have been modified by FEMA (e.g., latitudes and longitudes of locations rounded to the nearest 0.1°) due to confidentiality concerns. Metadata on each data field are provided from FEMA's Transaction Record Reporting and Processing (TRRP) documentation[40]. In order to construct an analog to USACE depth–damage functions, we filtered the data so that entries contained: total property value (actual cash value in whole dollars), total building damage (actual cash value in whole dollars; not insurance payout), and water depth (relative to the lowest occupied floor of the building, in feet). Entries for these fields

are derived from NFIP-certified loss adjusters' reports. Data points containing positive values for all three fields numbered 976,363. From this total, we focused on the most common building type: a one-story residential structure without a basement. Data fields including number of floors and basement type were sourced from the flood insurance application related to the claim. All data necessary for the analyses here were available for 493,707 flood losses to one-story residences without basements.

**Scenario-based analyses.** Analyses were performed under different scenarios so that different depth–damage relationships could be compared across different geographies, for different ages of building, for different values of building, when considering the flood management context when the claim was made and the building was constructed, and for specific historic flood events. When comparing across different geographies, the mean damage at each depth increment was sampled for each state and zip code, which are derived from the flood insurance applications. For different ages of building, vulnerability was split depending on the date of original construction or substantial improvement, whichever was more recent. For different values of building, stratification of vulnerability was guided by the total value of the building. When comparing flood management contexts, claims were split according to whether a building was situated in the FEMA 100-year flood zone and whether it was constructed before or after the publication of the map of this zone. These provide insights into whether, where, and how vulnerability differs across the US. Finally, for the event-based analysis, claims were extracted depending on whether the zip code fell within the considered USACE District and if the date of loss was during the considered event.

**US federal depth–damage curves.** We calculated the coefficient of determination (CoD; Eq. 1) to indicate the predictive power of US federal curves:

$$\text{CoD} = 1 - \frac{\sum_{n=1}^{N}\left(D_{\text{fed}}^{n} - D_{\text{nfip}}^{n}\right)^{2}}{\sum_{n=1}^{N}\left(D_{\text{nfip}}^{n} - \overline{D_{\text{nfip}}}\right)^{2}} \tag{1}$$

where $N$ is the number of claims, $D$ is damage, and the subscripts fed and nfip indicate whether the damage was estimated using federal curves or was from the empirical NFIP claims database respectively. If US federal curves were more informative than simply using the empirical mean, CoD > 0. Where it was not— that is, residual variance when using USACE curves exceeds the variance in the claims data itself—CoD < 0.

**Reporting summary.** Further information on research design is available in the Nature Research Reporting Summary linked to this article.

## Data availability
US depth–damage curves are available from the corresponding author upon request. A redacted version of the NFIP claims used in this analysis is available from ref. [25], though many of the fields necessary for replication of this study are missing. The full(er) database employed here is considered confidential, although enquiries for its availability may be sent to OpenFEMA@fema.dhs.gov.

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

## Acknowledgements
Oliver Wing and Paul Bates were supported by EPSRC grant EP/R511663/1. Paul Bates was also supported by a Leverhulme Research Fellowship and a Royal Society Wolfson Research Merit award.

## Author contributions
O.E.J.W. wrote the paper together with N.P., P.D.B., and C.K.; O.E.J.W. performed the analyses. All authors aided in the conceptualization of the research and interpretation of the results.

## Competing interests
The authors declare no competing interests.
