## [Peer Review File · Nature Communications]

Reviewers' comments:

Reviewer #1 (Remarks to the Author):

*General comments

This is an interesting, well-written paper where the authors explain that depth-damage curves are one of the principal sources of uncertainty in translation from hydraulic modelling through to flood losses. They highlight that the main assumptions around the use of depth-damage curves may be invalid and may mischaracterize flood vulnerability at the level of individual structures (through descriptive statistics of a large NFIP claims database). The authors thus propose an alternative to current depth-damage approaches, suggesting that damage can be better described by a beta function with bimodal distribution for different water depths. The suggested findings are intuitive and make sense, however the main finding (that NFIP flood losses are better characterized by a beta function) is not actually verified/validated (only visually described).

*Major comments

- The abstract suggests the main finding is that US depth-damage relationships are better described by a beta distribution than by standard monotonic curves. While this makes sense visually (looking at the data shown in Fig.2), the authors never formally validate their finding by comparing the beta and USACE curves (only USACE curves are tested, briefly, in Table S1). To be able to state in the abstract that "observed flood losses better follow a beta function" requires some justification of the beta distribution and (quantitative) validation.
- The description of the limitations of the current depth-damage approach is interesting and useful. However, if the main aim of the paper (l.60-68) is to test assumptions underpinning the depth-damage approach, then these assumptions must be stated upfront (and then tested). The manuscript should define its aims and pose clear research questions.
- The authors assess the predictive power of USACE depth-damage functions (l.470-472), but they do not do this for the proposed beta curves. If beta curves are to be preferred, this could/should be demonstrated... Also, please justify the choice of beta curves over other distributions.
- The main message of the manuscript remains a little unclear. The title "patterns in US flood vulnerability revealed from flood insurance big data" is unconvincing because (a) it is unclear what patterns are revealed, and (b) the expression "big data" is a bit of an oversell (I wouldn't call 2M rows "big data").
- The manuscript relies on a database that is not publicly accessible (l.139 "obtained directly from FEMA"). Please check data availability complies with Nature journals policy. I would also recommend stating the period of record upfront in the main text (currently it is only mentioned in the methods).

*Minor comments

- L.32. The term "flood depth" or "depth of inundation" is used throughout the manuscript; it might be helpful at some point to specify what this depth is relative to.
- L.112. "previous attempts to verify the reliability of depth damage curves": this sentence implies that the current paper attempts to verify their reliability, which is not really the case (it mostly describes NFIP patterns, and starts to look at the USACE curves in Table S1). Please remove "previous" to avoid confusion.
- L.116. This paragraph is a little confusing and could be written a little more clearly. It assumes a lot of prior knowledge on the part of the reader. Perhaps remove extraneous terms like "in isolation" (l.116) and try to clarify terms/expressions where possible (e.g. what are "different views of vulnerability" l.120? what is water "contamination" l.126?).
- L.139. Please be more specific about data accessibility. I thought this was the recently-released claims dataset until I read l.143.

- L.142. Define “all the information needed” and “additional explanatory fields necessary”
- L.151. Sentence is worth phrasing more clearly (“for depth values of 6”)
- L.168. Perhaps explain why most of the analysis is restricted to one-story buildings with no basement.
- L.200. Justify/quantify “relative flood damages... are better described by a beta distribution” statistically, or qualify the statement (this paragraph seems speculative otherwise)
- L.215 “could plausibly be the result of systematic geographical variation” – also a little speculative.
- L.226 “In order to test the regional variations in depth-damage across the US” – I don’t really see how this tests the regional variations. Is it possible to explain more explicitly here how the comparison shows e.g. the “characterization of the St Paul District as more vulnerable than the Chicago District is not justified” – where do you see this? I assume we are comparing Figs 1 and 3, but it is not obvious. It might help if you could make these comparisons more quantitative.
- Table S1. I really like this table; it’s quite informative and perhaps one of the most interesting/novel parts of the manuscript. However, these are just four events and I wonder if you could do something more systematic like consider a broader number of events and plot a scatterplot with error bars, to systematically assess the damage difference between NFIP/USACE? Alternatively, if you wanted to justify that the beta curves are a better approach than the USACE curves, couldn’t you do that here?
- L.269. Recommend referring to the figure, so reader knows where to look. The east-west divide is not obvious, and I wonder if this is due to the choice of colours.
- L.280. “current assumptions regarding simple geographical differences do not match the empirical data”: please clarify.
- L.311. Define FIRM on first use. The acronym is defined at line 329 but should be defined earlier in the text.
- L.370 “other sources of hydrologic “non-stationarity”” could be more explicit (what other sources).

*Figures

- Figure 1. Interesting visualisation, but I’m not sure this is a very fair “comparison” for those cities – wouldn’t it make more sense to show the NFIP data for each city alongside the USACE curve for the same city, rather than pooling all the NFIP data? Also, the different cities are a little hard to read – perhaps try increasing the font size and using thicker dashed lines if possible (e.g. Wilmington is hard to read). Perhaps you could display the boxplot medians in black and keep the colours for the dashed lines. Readability might also be a little clearer if you used something like a light shade of grey to colour in the boxes.
- Figure 2 is clear.
- Figure 3: the colours are difficult to read. I would recommend checking whether they are colourblind friendly and choosing a colour scale that better discriminates. Also, given your scale is null to positive, perhaps try using shades of one single colour (e.g. 0=white to 1=red), as the patterns might be clearer to read in doing so. If possible, try choosing a nice projection for the US map. I would also recommend including the period of record in the caption (here and on other figures, where relevant). Also, specify which dataset this is in the caption.
- Figure 4 is a little simple, like Figure 2, but otherwise fine. Again, it would be worth mentioning the period of record.
- Figure 5: the text for sample size could be larger. Not sure green/orange are best colours for colourblind readers? Not sure the main message of the figure is very clear?
- Figure S1. Seems a shame to show these curves here but not actually do anything with them. They are referred to twice in the main text, but only for visual comparison with Fig. 2., it seems.
- Figure S2. Worth explaining to the reader how you isolated these claims.

References

- The referencing is up-to-date but rather male-dominated.

Reviewer #2 (Remarks to the Author):

Review Criteria:

What are the major claims of the paper? Are they novel and will they be of interest to others in the community and the wider field? If the conclusions are not original, it would be helpful if you could provide relevant references. Is the work convincing, and if not, what further evidence would be required to strengthen the conclusions? On a more subjective note, do you feel that the paper will influence thinking in the field? Please feel free to raise any further questions and concerns about the paper.

Comments:

This study provides the results from a brief statistical analysis of the complete U.S. Federal Emergency Management Agency (FEMA) National Flood Insurance Program (NFIP) claims database. The authors evaluate the relative damages for one-story residential structures with no basement across several categorical variables, including binned depth of water, age of the structure, assessed structure value, and location (both geographic and inside/outside of the FEMA SFHA (the "100-year floodplain")). The authors' primary finding is that the observed depth-damage relationship in the claims database is poorly represented by the depth-damage curves used as the standard method for determining economic losses in flood risk studies. While not surprising, this finding is timely and relevant in light of on-going cost-benefit analyses for large structural projects aimed at reducing flood risk (e.g., Texas' 'Coastal Spine') and particularly relevant in discussions surrounding adaptation to increasing climate extremes. Many of the sub-findings in this study support those obtained through regional analysis of flood insurance claims (e.g., those in Brody et al.); however, this is the first study of its kind that has been conducted at a national scale and the findings are therefore particularly important to discussions surrounding flood risk management and the future of the NFIP.

It is my opinion that this article will promote a broader discussion around several hypothesis that are floating in the academic and professional communities about drivers of loss outside of floodplains and different mechanisms for flood risk management at a national scale (whether through the 100-year floodplain or another communication tool) and thus warrants publication.

The authors have provided sufficient information to reproduce the results from the study, however, it is important to note that the data utilized in this analysis is not publically available (due to PII concerns). The authors have also commented on this.

While I find that the manuscript is overall extremely well-written and has few gaps, I provide several suggestions for improvement below:

Lines 445-447 Please comment on whether the total building damage is the assessed damage or the NFIP payout. Note that insurance payouts are capped at \$250,000 for structural damage and using the payout data would impact the quality of the results. The authors should clearly state which data was used.

Figure 1 The authors do not discuss potential differences between freshwater and saltwater flooding and why these may drive regional differences in depth-damage relationships. This information is important in the context of Figure 1 where the USACE New Orleans depth-damage curve is much higher than any other curve (and also later in Figure 3). My recollection is that the NOLA curve was (re-) created after Hurricane Katrina using data from that event only and therefore (1) may represent significant saltwater flooding which is generally more damaging than freshwater and (2) may represent long-duration inundation which would explain the high relative damage for the bin of 1 foot inundation. While this does not have a significant impact on the discussion or findings of the study, it should probably be noted that the data used to create this curve may be anomalous to other studies. There are several points throughout the paper where this discussion might be relevant (for example regional differences, lines 269-281).

(Upon further review of the literature I believe that the background information for creation of the USACE New Orleans district curves can be found here:

<https://www.mvn.usace.army.mil/Portals/56/docs/PD/Donaldsv-Gulf.pdf>)

Lines 269-281 While the authors have grouped the data by state (and this is a reasonable jurisdictional assumption), it may more interesting (in the future) to look at differences by watershed. This should elucidate differences between flood hazards and, potentially, also differences in types of adaptation (elevation, wet-proofing/dry-proofing). I anticipate that there may large differences between coastal flood hazard and inland flood hazard areas.

Lines 307-383 Several previous studies that have also used the full FEMA dataset to conduct analyses of historical loss and attempted to understand what is driving flood damages both inside and outside of the FEMA floodplain as well as mitigation activities that reduce structural vulnerability to flooding (e.g., freeboard). While regional, the findings are supported by those herein and should be noted. Among others, these include:

Highfield W E and Brody S D 2013 Evaluating the Effectiveness of Local Mitigation Activities in Reducing Flood Losses *Nat. Hazards Rev.* 14 229–36

Highfield W E, Norman S A and Brody S D 2013 Examining the 100-Year Floodplain as a Metric of Risk, Loss, and Household Adjustment *Risk Anal.* 33 186–91

Brody S D, Sebastian A, Blessing R and Bedient P B 2015 Case study results from southeast Houston, Texas: identifying the impacts of residential location on flood risk and loss *J. Flood Risk Manag.* n/a-n/a Online: <http://doi.wiley.com/10.1111/jfr3.12184>

Brody S D, Blessing R, Sebastian A and Bedient P B 2013 Delineating the Reality of Flood Risk and Loss in Southeast Texas *Nat. Hazards Rev.* 14 89–97

Line 394 For clarity, the authors might add the word “where” prior to “each”

Reviewer #3 (Remarks to the Author):

A very interesting manuscript. I have not seen such analysis on such a large set of NFIP claims prior to this.

It was very interesting and well researched paper. I don't have much critique. Your methods and analysis were clearly demonstrated and communicated as well as your findings in the paper.

Its' a very interesting piece of work and I think is very informative regarding the use and application of depth-damage curves and how they compare to NFIP claims data.

Reviewer #1

*General comments

This is an interesting, well-written paper where the authors explain that depth-damage curves are one of the principal sources of uncertainty in translation from hydraulic modelling through to flood losses. They highlight that the main assumptions around the use of depth-damage curves may be invalid and may mischaracterize flood vulnerability at the level of individual structures (through descriptive statistics of a large NFIP claims database). The authors thus propose an alternative to current depth-damage approaches, suggesting that damage can be better described by a beta function with bimodal distribution for different water depths. The suggested findings are intuitive and make sense, however the main finding (that NFIP flood losses are better characterized by a beta function) is not actually verified/validated (only visually described).

We firstly thank this reviewer for a thorough and perceptive examination of our work. We are very grateful that they recognise the value of such an analysis and have offered constructive feedback in order to ensure maximal scientific robustness of our claims.

*Major comments

- The abstract suggests the main finding is that US depth-damage relationships are better described by a beta distribution than by standard monotonic curves. While this makes sense visually (looking at the data shown in Fig.2), the authors never formally validate their finding by comparing the beta and USACE curves (only USACE curves are tested, briefly, in Table S1). To be able to state in the abstract that “observed flood losses better follow a beta function” requires some justification of the beta distribution and (quantitative) validation.

Thank you for raising this. Our primary intention in the presentation of a beta distribution is to aid the interpretation of the shape of the depth-damage distribution: namely, concentration at minimal and maximal modes for a given depth. If we intended to produce a “best fit distribution”, we would simply have used the empirical one. Given that the entire sample of claims data contribute to the fitting of the beta distribution, there is no “validation” possible; it is just a tool by which to formulaically describe the distributions shown in Fig. 2.

We do take the point in the spirit with which it was intended however, and analyse how well the beta distributions were fitted to the data (the closest thing to validation possible) and compare these to how well a single deterministic value (the federal curves) fit. See Supplementary Note 1, Figs. S2, S3 and Table S1. The R^2 values are mostly extremely high, and in all cases exceed the deterministic case. Hopefully, this now justifies our comment that “observed flood losses better follow a beta function”.

In the table below, we also show a split-sample (pseudo-)validation procedure whereby we fit separate beta distributions to two random samples of 50% of the claims data, demonstrating the stability of beta parameter estimates. In brackets are the 95% confidence bounds.

Depth increment (feet)	α		β	
	Sample 1	Sample 2	Sample 1	Sample 2
1	0.423 (0.418, 0.428)	0.422 (0.417, 0.427)	0.801 (0.799, 0.804)	0.805 (0.802, 0.807)

2	0.484 (0.476, 0.493)	0.480 (0.472, 0.489)	0.658 (0.653, 0.662)	0.646 (0.641, 0.651)
3	0.491 (0.479, 0.503)	0.491 (0.479, 0.503)	0.525 (0.519, 0.531)	0.520 (0.513, 0.526)
4	0.534 (0.519, 0.549)	0.525 (0.511, 0.540)	0.413 (0.405, 0.421)	0.411 (0.403, 0.419)
5	0.686 (0.665, 0.708)	0.667 (0.647, 0.689)	0.428 (0.414, 0.442)	0.423 (0.409, 0.437)
7	0.802 (0.770, 0.835)	0.791 (0.761, 0.823)	0.386 (0.368, 0.404)	0.376 (0.358, 0.395)

We cannot validate these beta distributions in the same way as the federal curves because they do not produce a single building-level damage value, but a spread. Indeed, this is the point of the paper: that flood damages are inherently unpredictable and so should be treated stochastically. All we would uncover for a given claim is: “yes, the recorded damage lies within the spread of the probabilistically modelled (based on the beta distributions) damage value” (which, of course, would be any value between 0 and 100% damage). Our (more thorough) examination of the federal curves is to demonstrate the fallacy of a one-to-one depth-damage relationship and to highlight their incredibly low skill in building-level damage estimation in spite of their universal use. To avoid criticisms of brevity in this strand of analysis, we extend the federal validation by adopting the widely used FIA curve to predict all building losses in the NFIP database (residential, one story, no basement). This bolsters the conclusions drawn from looking at the relative depth-damage curves (Fig. 1) with real absolute damages. The results are shown in Table S2, where low-depth damages are overestimated by roughly a quarter and high-depth damages are underestimated by roughly a quarter *in aggregate*. Across all depths, *at the building level*, mean (absolute) errors from the FIA curve are 84% of the mean damage. The negative coefficients of determination indicate low FIA curve skill across the board.

- The description of the limitations of the current depth-damage approach is interesting and useful. However, if the main aim of the paper (1.60-68) is to test assumptions underpinning the depth-damage approach, then these assumptions must be stated upfront (and then tested). The manuscript should define its aims and pose clear research questions.

Thanks for raising this. We more clearly define the aims and scope of our research, which are teed up following the review section (thus the research questions follow this section; lines 132-140).

- The authors assess the predictive power of USACE depth-damage functions (1.470-472), but they do not do this for the proposed beta curves. If beta curves are to be preferred, this could/should be demonstrated.... Also, please justify the choice of beta curves over other distributions.

This has been addressed above (Supplementary Note 1, Figs. S2, S3 and Table S1). The beta distribution is preferred to others because: (i) it can represent the distinct bimodal shape shown in Fig. 2; and (ii) it is defined in the interval [0,1], as relative damages are (other distributions would have $\text{cdf} > 0$ when relative damage ≤ 0 and $\text{cdf} < 1$ when relative damage ≥ 1 , which are impossible). We illuminate this in Supplementary Note 1.

- The main message of the manuscript remains a little unclear. The title “patterns in US flood vulnerability revealed from flood insurance big data” is unconvincing because (a) it is unclear what patterns are revealed, and (b) the expression “big data” is a bit of an oversell (I wouldn’t call 2M rows “big data”).

Thanks for raising this. We agree that perhaps the word “patterns” does not convey the nature and volume of analysis performed (and perhaps only refers to Fig. 3). As such, we exchange this for the word “insights”. The insights we reveal are: the empirical depth-damage relationship (Fig. 2) and its formulaic description (Fig. S1), its spatial distribution (Fig. 3), the validity of a relative function (Fig. 4), the effect of floodplain management (Fig. 5), and that current operational practice misestimates depth-damage (Fig. 1, S2, S3, S4; Table S1, S2, S3). Hopefully textual changes throughout the manuscript have strengthened these messages.

For (b), I suppose “big” is in the eye of the beholder. In the context of flood loss assessments, this is very much “big data” territory. We analyze structural flood vulnerability with a volume of data that is without precedent: previous works have always been hamstrung by data availability and commonly use on the order 10^2 – 10^3 data points. We believe the “big data” label sufficiently conveys the contextual magnitude of this study.

- The manuscript relies on a database that is not publicly accessible (l.139 “obtained directly from FEMA”). Please check data availability complies with Nature journals policy. I would also recommend stating the period of record upfront in the main text (currently it is only mentioned in the methods).

Indeed, it is a pity that this is not fully publicly available – and we use this paper as a platform to pressure FEMA into releasing it. Data confidentiality does not preclude publication of what we believe to be important conclusions, however. We also state the time period up front as suggested (line 145).

*Minor comments

- L.32. The term “flood depth” or “depth of inundation” is used throughout the manuscript; it might be helpful at some point to specify what this depth is relative to.

Thanks. We make this clearer throughout the manuscript (line 146; 477).

- L.112. “previous attempts to verify the reliability of depth damage curves”: this sentence implies that the current paper attempts to verify their reliability, which is not really the case (it mostly describes NFIP patterns, and starts to look at the USACE curves in Table S1). Please remove “previous” to avoid confusion.

This is a point upon which we must regrettably disagree. This paper very much verifies the reliability of depth-damage curves (Fig. 1, S2, S3, S4; Table S1, S2, S3), finding ubiquitous federal curves match poorly with depth-damage observations. Indeed, this is one of the main conclusions of the paper.

- L.116. This paragraph is a little confusing and could be written a little more clearly. It assumes a lot of prior knowledge on the part of the reader. Perhaps remove extraneous terms like “in isolation” (l.116) and try to clarify terms/expressions where possible (e.g. what are “different views of vulnerability” l.120? what is water “contamination” l.126?).

Thank you for raising this. This paragraph has been altered to ensure that it can be understood by the journal’s broad readership (lines 112-130).

- L.139. Please be more specific about data accessibility. I thought this was the recently-released claims dataset until I read l.143.

Thanks. We make clear that the data are confidential in a data availability statement at the end.

- L.142. Define “all the information needed” and “additional explanatory fields necessary”

Thanks. We define this here and refer the reader to the methods section for further details.

- L.151. Sentence is worth phrasing more clearly (“for depth values of 6”)

Thanks. We have done so.

- L.168. Perhaps explain why most of the analysis is restricted to one-story buildings with no basement.

We state that this building type constitutes a majority of claims and further describe the rationale by keeping our analysis focused on this in lines 177-178.

- L.200. Justify/quantify “relative flood damages... are better described by a beta distribution” statistically, or qualify the statement (this paragraph seems speculative otherwise)

We flesh out the description of results here in light of previous points made: namely, that we justify the beta distribution’s skill over the standard federal curves (lines 218-222).

- L.215 “could plausibly be the result of systematic geographical variation” – also a little speculative.

Given it is the first sentence of a new section, its intention is to speculatively tee up a new hypothesis which we then go on to test. We clarify this further (line 245).

- L.226 “In order to test the regional variations in depth-damage across the US” – I don’t really see how this tests the regional variations. Is it possible to explain more explicitly here how the comparison shows e.g. the “characterization of the St Paul District as more vulnerable than the Chicago District is not justified” – where do you see this? I assume we are comparing Figs 1 and 3, but it is not obvious. It might help if you could make these comparisons more quantitative.

Apologies for the lack of clarity in this passage. We add quantifications to these results to aid interpretation and ensure the reader is aware Figs. 1 and 3 are to be compared (lines 256-270). The two points we are making: (i) the differences in damage at 1-foot inundation between different USACE curves are unsupported by the data (e.g. St. Paul = 32% (USACE) & 16% (NFIP); Chicago = 16% (USACE) & 24% (NFIP)) and (ii) more aggregate (state/district level) summaries of vulnerability often mask considerable variation at the zip code level. Hopefully this is now clear.

- Table S1. I really like this table; it’s quite informative and perhaps one of the most interesting/novel parts of the manuscript. However, these are just four events and I wonder if you could do something more systematic like consider a broader number of events and plot a scatterplot with error bars, to systematically assess the damage difference between

NFIP/USACE? Alternatively, if you wanted to justify that the beta curves are a better approach than the USACE curves, couldn't you do that here?

Thanks for raising these points. Hopefully, most of them have been answered elsewhere in this response. The reason this strand of analysis is relatively limited is that we needed to isolate events specific to the USACE district to examine if they have skill in the locality in which they were developed. Broader national scale analyses with the “national” FIA curve have now been executed. (Table S2).

- L.269. Recommend referring to the figure, so reader knows where to look. The east-west divide is not obvious, and I wonder if this is due to the choice of colours.

Thanks, have done so (line 298). The concentrations of oranges and yellows in the west differs contrasts to the greater prevalence of greens in the east.

- L.280. “current assumptions regarding simple geographical differences do not match the empirical data”: please clarify.

Thanks, have done so (line 314).

- L.311. Define FIRM on first use. The acronym is defined at line 329 but should be defined earlier in the text.

Thanks, we remove the term here and retain line 329 (now line 363) as the first use.

- L.370 “other sources of hydrologic “non-stationarity”” could be more explicit (what other sources).

We elucidate the main sources of non-stationarity (lines 403-405) but, with no need to be exhaustive, we retain the “other...” clause.

*Figures

- Figure 1. Interesting visualisation, but I'm not sure this is a very fair “comparison” for those cities – wouldn't it make more sense to show the NFIP data for each city alongside the USACE curve for the same city, rather than pooling all the NFIP data? Also, the different cities are a little hard to read – perhaps try increasing the font size and using thicker dashed lines if possible (e.g. Wilmington is hard to read). Perhaps you could display the boxplot medians in black and keep the colours for the dashed lines. Readability might also be a little clearer if you used something like a light shade of grey to colour in the boxes.

We believe that the comparison is fair. These are not cities, but USACE districts which can be much broader (e.g. the Wilmington district covers North Carolina). More importantly, however, is that these are simply the sources of these data: their applications spill far beyond the geographical boundaries of the district (e.g. Galveston curves are commonly used nationally). The regional comparisons the reviewer suggests appear in a separate section (around Fig. 3).

Thanks for the comments on legibility – we have acted upon these.

- Figure 2 is clear.

Thanks.

- Figure 3: the colours are difficult to read. I would recommend checking whether they are colourblind friendly and choosing a colour scale that better discriminates. Also, given your scale is null to positive, perhaps try using shades of one single colour (e.g. 0=white to 1=red), as the patterns might be clearer to read in doing so. If possible, try choosing a nice projection for the US map. I would also recommend including the period of record in the caption (here and on other figures, where relevant). Also, specify which dataset this is in the caption.

Thanks for this feedback. We want to keep both A and B on the same scale to emphasize points made in the text and, given the variability at the zip code level, this necessarily homogenizes the state level map somewhat (it becomes less discriminatory). Shades of a single colour necessitate data pertaining to the colour white, which then blends in with the “nodata” background of the zip code map and obviously the rest of the page too. Orange and green are selected as a colour scale which does not pass through white and is readily discriminatory to those with colour blindness. Indeed, all of our colours were selected with this in mind based on the works of Okabe & Ito (2002; <https://jfly.uni-koeln.de/color/>) but thanks for highlighting this. We see little point in consistently repeating the period of record in the figures as this remains unchanged, but we do specify the dataset as requested.

- Figure 4 is a little simple, like Figure 2, but otherwise fine. Again, it would be worth mentioning the period of record.

Thanks. Again, we see little need to repeat the period of record.

- Figure 5: the text for sample size could be larger. Not sure green/orange are best colours for colourblind readers? Not sure the main message of the figure is very clear?

We increase the legibility of the sample sizes as requested. As per Okabe & Ito (2002), the selected colours are colour-blind safe. It is a complex message which we are trying to convey, and we can do nothing more than present the data. Hopefully our interpretations of this figure in the text (lines 371-385) aid the reader in understanding the main message.

- Figure S1. Seems a shame to show these curves here but not actually do anything with them. They are referred to twice in the main text, but only for visual comparison with Fig. 2., it seems.

Thanks. We believe this point has been addressed elsewhere.

- Figure S2. Worth explaining to the reader how you isolated these claims.

Thanks, we now do so.

References

- The referencing is up-to-date but rather male-dominated.

Thanks for flagging this and we appreciate the sentiment. However, we were unaware of the gender balance of the citation list simply because there is no way for us to check without personal knowledge of the author team on each paper. We selected the most relevant

references to this work amidst gender-blindness. If there are any prominent female-led works that are missing from our discussions, we would gladly be pointed towards them.

Reviewer #2

Review Criteria:

What are the major claims of the paper? Are they novel and will they be of interest to others in the community and the wider field? If the conclusions are not original, it would be helpful if you could provide relevant references. Is the work convincing, and if not, what further evidence would be required to strengthen the conclusions? On a more subjective note, do you feel that the paper will influence thinking in the field? Please feel free to raise any further questions and concerns about the paper.

Comments:

This study provides the results from a brief statistical analysis of the complete U.S. Federal Emergency Management Agency (FEMA) National Flood Insurance Program (NFIP) claims database. The authors evaluate the relative damages for one-story residential structures with no basement across several categorical variables, including binned depth of water, age of the structure, assessed structure value, and location (both geographic and inside/outside of the FEMA SFHA (the “100-year floodplain”). The authors’ primary finding is that the observed depth-damage relationship in the claims database is poorly represented by the depth-damage curves used as the standard method for determining economic losses in flood risk studies. While not surprising, this finding is timely and relevant in light of on-going cost-benefit analyses for large structural projects aimed at reducing flood risk (e.g., Texas’ ‘Coastal Spine’) and particularly relevant in discussions surrounding adaptation to increasing climate extremes. Many of the sub-findings in this study support those obtained through regional analysis of flood insurance claims (e.g., those in Brody et al.); however, this is the first study of its kind that has been conducted at a national scale and the findings are therefore particularly important to discussions surrounding flood risk management and the future of the NFIP.

It is my opinion that this article will promote a broader discussion around several hypothesis that are floating in the academic and professional communities about drivers of loss outside of floodplains and different mechanisms for flood risk management at a national scale (whether through the 100-year floodplain or another communication tool) and thus warrants publication.

The authors have provided sufficient information to reproduce the results from the study, however, it is important to note that the data utilized in this analysis is not publically available (due to PII concerns). The authors have also commented on this.

While I find that the manuscript is overall extremely well-written and has few gaps, I provide several suggestions for improvement below:

We are incredibly thankful of this reviewer’s kind remarks on our research. We respond to each of their suggestions for improvement below.

Lines 445-447 Please comment on whether the total building damage is the assessed damage or the NFIP payout. Note that insurance payouts are capped at \$250,000 for structural

damage and using the payout data would impact the quality of the results. The authors should clearly state which data was used.

Thanks for raising this. We clarify in the methods section that the damage values we employ are the assessed damages and unrelated to the subsequent payout (line 484).

Figure 1 The authors do not discuss potential differences between freshwater and saltwater flooding and why these may drive regional differences in depth-damage relationships. This information is important in the context of Figure 1 where the USACE New Orleans depth-damage curve is much higher than any other curve (and also later in Figure 3). My recollection is that the NOLA curve was (re-) created after Hurricane Katrina using data from that event only and therefore (1) may represent significant saltwater flooding which is generally more damaging than freshwater and (2) may represent long-duration inundation which would explain the high relative damage for the bin of 1 foot inundation. While this does not have a significant impact on the discussion or findings of the study, it should probably be noted that the data used to create this curve may be anomalous to other studies. There are several points throughout the paper where this discussion might be relevant (for example regional differences, lines 269-281).

(Upon further review of the literature I believe that the background information for creation of the USACE New Orleans district curves can be found here:

<https://www.mvn.usace.army.mil/Portals/56/docs/PD/Donaldsv-Gulf.pdf>)

Thanks, yes, this is an important point to raise. In Figure 1, we selected *only* curves which represent freshwater flooding. We make this clearer (lines 103; 105; 172). Often coastal Corps districts have separate curves for salt/freshwater flooding, as is the case with New Orleans. However, the selected NOLA curve was for “long duration” freshwater flooding so, for balance, we also place the “short duration” curve in the figure. As the reviewer suggests, this does not impact on our discussion or findings.

We further discuss the impact of salt vs fresh water flooding in the regional differences section (lines 303-307). Although some flood claims had an indication of flood source, their relative damages did not chime with our physical understanding (that coastal floods are more damaging than fluvial floods for a given depth). See Fig. S5 for a demonstration of this.

Lines 269-281 While the authors have grouped the data by state (and this is a reasonable jurisdictional assumption), it may more interesting (in the future) to look at differences by watershed This should elucidate differences between flood hazards and, potentially, also differences in types of adaptation (elevation, wet-proofing/dry-proofing). I anticipate that there may large differences between coastal flood hazard and inland flood hazard areas.

We agree that this would be an interesting direction for future research. This is somewhat elucidated by Fig. 3B where claims are grouped by zip code; though data coarseness in the national database can prove a problem in assigning claims to a specific watershed.

Lines 307-383 Several previous studies that have also used the full FEMA dataset to conduct analyses of historical loss and attempted to understand what is driving flood damages both inside and outside of the FEMA floodplain as well as mitigation activities that reduce structural vulnerability to flooding (e.g., freeboard). While regional, the findings are supported by those herein and should be noted.

Among others, these include:

Highfield W E and Brody S D 2013 Evaluating the Effectiveness of Local Mitigation Activities in Reducing Flood Losses Nat. Hazards Rev. 14 229–36
Highfield W E, Norman S A and Brody S D 2013 Examining the 100-Year Floodplain as a Metric of Risk, Loss, and Household Adjustment Risk Anal. 33 186–91
Brody S D, Sebastian A, Blessing R and Bedient P B 2015 Case study results from southeast Houston, Texas: identifying the impacts of residential location on flood risk and loss J. Flood Risk Manag. n/a-n/a Online: <http://doi.wiley.com/10.1111/jfr3.12184>
Brody S D, Blessing R, Sebastian A and Bedient P B 2013 Delineating the Reality of Flood Risk and Loss in Southeast Texas Nat. Hazards Rev. 14 89–97

Thanks for raising this. We flesh out our discussions with these regional analyses in lines 400-414. It should be noted that the cited studies focus more broadly on flood *risk*, while our research truly isolates flood *vulnerability* (independent of the hazard and exposure).

Line 394 For clarity, the authors might add the word “where” prior to “each”

Thanks, we do so.

Reviewer #3

A very interesting manuscript. I have not seen such analysis on such a large set of NFIP claims prior to this.

It was very interesting and well researched paper. I don't have much critique. Your methods and analysis were clearly demonstrated and communicated as well as your findings in the paper.

Its' a very interesting piece of work and I think is very informative regarding the use and application of depth-damage curves and how they compare to NFIP claims data.

Thanks to this reviewer for their kind comments.

REVIEWERS' COMMENTS:

Reviewer #1 (Remarks to the Author):

The authors have made considerable efforts to strengthen and enhance the paper. In the revised manuscript, they have added additional information illustrating the fit of the beta distributions and compared this to the fit of the federal curves. These supplementary analyses and figures are quite impactful.

I also find the additional analyses that extend the federal validation to all building losses in the NFIP database worthwhile.

The aims, title, and main messages have all been clarified, and all other points have been addressed in detail.

I agree that their paper now demonstrates "the fallacy of a one-to-one depth-damage relationship" and assesses the reliability of depth-damage curves quite convincingly.

This is an interesting analysis, and I wholeheartedly recommend publication.

Reviewer #2 (Remarks to the Author):

The authors have sufficiently addressed my previous comments.